# Based on abnormal fluctuations in user-side flow simulation analysis of low- and medium-pressure gas pipeline leakage monitoring

**Xiaomin Wang** [iD] *[⊕], **Zhengshan Luo**[⊕], **Yulei Kong, Qingqing Wang**

School of Management, Xi'an University of Architecture and Technology, Xi'an, Shaanxi, China

⊕ These authors contributed equally to this work.
* wxm@xauat.edu.cn

## Abstract

Due to the weak monitoring equipment for low- and medium-pressure gas pipelines, it is not easy to identify small flow leaks. The detection methods are mostly traditional manual inspections or night pressure-maintaining leak detection methods, which cannot be automatically monitored and sensed immediately. The abnormal fluctuations in client traffic caused by pipeline leaks studied in this paper can locate and detect leak locations more effectively. This paper analyzes the theoretical formula of leakage location based on flow data and finds out how to use the abnormal fluctuation of user-side flow to detect and locate gas pipeline leakage. First of all, this article uses the simulation software Pipeline Studio to construct the medium and low-pressure pipeline model. On this basis, 6 sets of leakage conditions were designed and simulated dynamically. Finally, the simulation results are analyzed, and the results show that: 1) The advisable monitoring period for monitoring abnormal fluctuations of user-side traffic is 10s. 2) There are two relationships between abnormal flow fluctuations and leakage position. They are: when the leakage point is at the first 40% of the relative distance from the gas source, the disturbance amplitude first increases and then decreases, and at the last 60%, it continues to decrease.; The closer the leak is to the user end, the more significant the abnormal flow fluctuations will be. On the contrary, the smaller the abnormal flow fluctuations will be; 3) No matter where the leakage occurs, the abnormal flow fluctuations in the 2nd and 3rd seconds after the leak occurs tend to be consistent. The proposal of the advisable monitoring period and the relationship between abnormal fluctuations of flow and the location of leakage provides a theoretical basis for the use of abnormal fluctuations of user-side flow for gas pipeline leakage detection and location.

## Introduction

With the increasing service life of gas pipelines, the gas pipelines put into operation in the early stage are gradually aging [1]. Pipe anti-corrosion layer falls off and then on the erosion wear and tear, wall thickness and decreases, pipeline and weld corrosion perforation and rupture, at the same time, also due to short service life of gas pipeline third-party damage pipeline

**Data Availability Statement:** All relevant data are within the paper and its Supporting Information files.

**Funding:** Prof ZS Luo, one of the authors of this manuscript, received these funds including the

National Natural Science Foundation of China (41877527) and Shaanxi Social Science Fund Project (2018S34). His work in this draft includes methodology, data curation, visualization, writing-original draft preparation.

**Competing interests:** The authors have declared that no competing interests exist.

network system construction, the natural disaster, the other, the influence of such factors as prone to accidental removal of pipe coating, high-pressure erosion sex perforation, and fracture. Gas pipeline leak accidents not only cause waste of resources and pollution but also easily cause casualties. Therefore, after the occurrence of gas pipeline leak accidents, how to efficiently and accurately detect and locate the gas pipeline is a key topic for relevant scholars at home and abroad.

It is easy to cause a gas leak in the middle and low-pressure gas pipeline due to thin pipe walls, poor anticorrosion coating quality, lack of cathodic protection, and external damage to the construction. In Xi'an, for example, low and medium pressure gas pipelines account for less than 60% of all gas pipelines, but the proportion of leak incidents is more than 80%. Moreover, as the low and medium pressure pipelines are closer to buildings, the damage caused by leaks and explosions will be more serious. However, the monitoring equipment equipped with middle and low-pressure gas pipelines is weak, and it is difficult to identify small flow leaks. The detection methods are mostly traditional manual inspection or night pressure maintenance, which cannot be automatically monitored or instantly sensed. For whole promoting the comprehensive diagnostic capacity of gas pipeline network, the implementation of low-pressure gas pipeline leak perception function, this paper takes the low-pressure pipeline as the research object, through the research and application of the theory of leak through the data analysis, to establish gas pipeline leak simulation test system in the laboratory, summarize laws, to establish the mathematical model of a pipeline leak, A judgment method is proposed for leak detection of middle and low-pressure pipelines.

At present, leak detection and location technologies for gas pipelines at home and abroad are mainly divided into two categories, namely direct detection method and indirect detection method [2]. Direct detection methods include manual inspection method [3], infrared imaging method [4], optical fiber sensor detection method [5], pipeline robot method [6], etc. Indirect detection methods include the pressure gradient method [7], negative pressure wave method [8], mass flow balance method, and statistical analysis method [9]. However, each leak detection technology has its own advantages and disadvantages. The biggest advantage of the direct detection method is its high detection and positioning accuracy, but it requires an additional investment of expensive manpower or equipment, which is often not acceptable to gas enterprises. In the indirect detection method [10], the negative pressure wave propagates and attenuates too fast in the gas phase [11], which is not conducive to leak location and detection. Statistical analysis and genetic formula require a large number of historical data [12], and there is inevitable random error in positioning accuracy. The mass flow balance method and pressure gradient method need to construct complex mathematical models [13]. Therefore, it is not good to use a single method for leak detection and location. Domestic and foreign scholars, such as Pal-Stefan Murvay et al. [14], summarized the gas leak detection and positioning technology and pointed out that a hybrid detection and positioning system should be formed by combining various methods according to the actual situation. Morteza Zadkarami et al. [15] compared the single leak detection method and the fusion leak detection method and concluded that the fusion statistical characteristics and wavelet characteristics of the leak detection method [16] are more stable than the single use of a certain method. Yang S et al. [17] used a UAV equipped with a methane detector and transmitted the detection results in real-time to form the optimal raster scanning strategy, which greatly improved the efficiency of UAV inspection. Compared to the leak characteristics, the user side pressure and flow data are easier to monitor, and there are few studies on pipeline leak detection and location based on user side flow data yet. Therefore, this paper intends to detect and locate gas pipeline leak by using the abnormal fluctuation of flow on the user side generated at the moment of middle and low-pressure gas pipeline leak combined with other detection technologies under different working

conditions, which provides an active basis theory, method, and conclusion for the low-pressure pipeline leak detection and location work.

This paper analyzes the theoretical formula of leakage location based on the flow data and concludes that using the abnormal fluctuation of the flow to replace the flow data has a better application effect. Then use the simulation software Pipeline Studio to build the medium and low-pressure pipeline model, and design 6 sets of leakage conditions for dynamic simulation. Finally, the simulation results are analyzed. The analyses reveal the advisable monitoring period for abnormal fluctuations of user-side flow and the relationships between the abnormal flow fluctuation and the leakage position. This article provides a theoretical basis for the detection and location of the leakage of medium and low-pressure gas pipelines by using the abnormal fluctuation of the user-side flow, draws meaningful conclusions, and proposes an effective judgment method for the leakage detection of the medium and low-pressure gas pipelines.

The following chapters are outlined as follows: In Chapter 2, the theoretical formula of leak location based on flow data is firstly analyzed mathematically, and it is found that using abnormal fluctuation of flow data to replace flow data has a better application effect in pipeline leak detection and location, which is also the theoretical basis and premise of this study. Then, the simulation software Pipeline Studio is used to build the low and medium pressure Pipeline model. On this basis, six groups of leak conditions were designed and dynamic simulation was carried out, and the corresponding simulation results were obtained. Chapter 3 discusses the advisable monitoring period of abnormal fluctuation monitoring of user side traffic based on simulation results. At the same time, the relationship between the abnormal fluctuation of flow and leak location is quantitatively explored. Chapter 4 and 5 give the main conclusions of this study and further discussion, respectively.

## Related work

### Theoretical formula of gas flow at leak point of pipeline

When hole leak occurs in straight round pipe, the leak process is considered as adiabatic process. Assuming that indoor gas is an ideal gas, Bernoulli equation and adiabatic equation are used to deduce the leak velocity algorithm of the leak gas in the pipeline, as shown in Eq 1.

$$V = \sqrt{\frac{2K}{K-1}RT[1 - (\frac{P_0}{P_1})^{\frac{K-1}{K}}]} \tag{1}$$

Where, V refers to the leak velocity of the leak gas in the pipeline, m/s; K is isentropic index, and 1.3 is used for gas. T refers to the real-time temperature of natural gas, K; P0 is the atmospheric pressure, Pa; $P_1$ is the pressure at the pipe head, Pa.

After the modification of the flow coefficient at the orifice, the velocity calculation relation of the leak gas is transformed into Formula (2).

$$Q_v = C_D A_\varphi \sqrt{\frac{2K}{K-1}RT[1 - (\frac{P_0}{P_1})^{\frac{K-1}{K}}]} \tag{2}$$

In the formula, $\varphi$ represents the velocity coefficient under different states, which is 0.97~0.98 under general conditions. $Q_v$ is the theoretical volume flow rate of the leak port, m3/s; $A_\varphi$ represents the equivalent area of the leak outlet, which is generally the same as the measured area, m2; R is the gas constant.

When leak occurs, the theoretical flow rate at the leak point actually depends on the difference of internal and external pressure at the leak point and the form of leak hole. The leak of a pressure vessel is equivalent to the flow of the medium in the vessel from one pressure

condition to another pressure condition, while the other pressure condition when the gas pipeline leaks is generally the ambient atmosphere, so the formula of gas mass flow at the leak place [18] is shown below.

$$Q_m = A_e(P_0 + P_2)\sqrt{\frac{k}{RT_0}(\frac{2}{k+1})^{\frac{k+1}{2(k-1)}}} \tag{3}$$

Where, $Q_m$ is the mass flow rate of gas at the leak place, kg/s; $A_g$ is the actual flow area of the leak hole, m2; $P_2$ is the pressure in the pipe at the leak point, kPa; $T_0$ is the standard temperature, K.

The calculation formula of medium and low-pressure gas pipelines in China generally adopts the formula recommended in "Urban Gas Design Code" [19], and gradually forms the current general calculation formula, as shown in Eq 4.

$$\frac{P_1{}^2 - P_2{}^2}{L} = 1.27 \times 10^7 \lambda \frac{\rho Q^2}{D^5} \frac{T_f}{T_0} Z \tag{4}$$

Where, L is the length of the pipeline, m; $\lambda$ is the friction resistance coefficient of the pipeline; $\rho$ is the gas density, kg/m3; D is the pipe diameter, mm; Q is the gas flow in the pipeline, m3/h; $T_f$ is the gas temperature, K; Z is the compression factor.

According to simultaneous Formulas (3) and (4), different leak locations of gas pipelines will lead to different values of $P_2$, thus changing the value of $Q_m$. Therefore, in the case of known $Q_m$, leak point $P_2$ can be reversely derived, and then the value of L can be derived to realize leak detection and location using flow data.

## Construct a pipe network model (PNM)

In this simulation, the set of users, pipelines, and nodes is based on the actual operation data of the middle and low-pressure gas pipeline systems (Fig 1). There are three users in the model, namely, gas source user (1), user (2), and leak point (3). The set gas pipeline (A) only includes

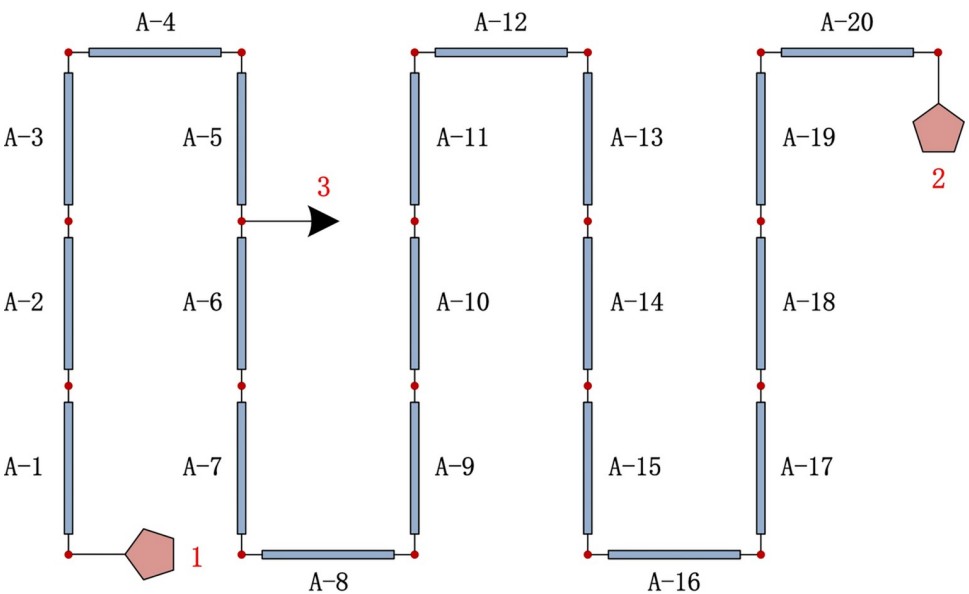

**Fig 1. Pipe network model.**

a straight pipe with a length of 400m. Then divide the pipe evenly into 20 sections, each of which has the same length. As a result, there will be 19 nodes. In the figure, leak point 3 is at node F between A-5 and A-6 to simulate leak conditions at different leak locations. The parameters and numbers of pipes and nodes are shown in Table 1.

## Simulations of pipeline leak

**Assumptions.** Based on the pipe network model constructed in the subsection 'Construct a pipe network model (PNM)', this section simulates the gas flow of the user side with a set leak situation. It should be noted that many affecting factors are involved in the research of middle and low-pressure gas pipeline leak. However, all relevant physical factors cannot be taken into full consideration in the laboratory simulation. Therefore, we simplify the model appropriately to reduce the complexity of the research problem. In order to ensure the scientific of the simplified model, the following assumptions [20] are made in this paper.

1. Natural gas is assumed to be an ideal gas in a compressible state, and follows the gas equation and law in the ideal state during the flow process in the tube.

2. The heat transfer reaction between air and gas in the whole diffusion process is ignored, so the changes in physical quantities such as airflow velocity caused by heat transfer need not be considered [21].

3. The friction resistance at the leak outlet is set to be constant in the process of gas leak and does not change with the change of flow rate and external conditions [22].

**Leak conditions.** The gas flow rate on the user side is related to many factors, including the gas supply and pressure of the gas source user, the gas supply form (constant gas supply

**Table 1. Pipe and node parameters and numbers.**

| Pipeline ID | Length (m) | Internal diameter (mm) | Wall thickness (mm) | Absolute equivalent roughness (mm) | Node ID | |
|---|---|---|---|---|---|---|
| A-1 | 20 | 106 | 6 | 0.025 | Node B | |
| A-2 | 20 | 106 | 6 | 0.025 | | Node C |
| A-3 | 20 | 106 | 6 | 0.025 | Node D | |
| A-4 | 20 | 106 | 6 | 0.025 | | Node E |
| A-5 | 20 | 106 | 6 | 0.025 | Node F | |
| A-6 | 20 | 106 | 6 | 0.025 | | Node G |
| A-7 | 20 | 106 | 6 | 0.025 | Node H | |
| A-8 | 20 | 106 | 6 | 0.025 | | Node I |
| A-9 | 20 | 106 | 6 | 0.025 | Node J | |
| A-10 | 20 | 106 | 6 | 0.025 | | Node K |
| A-11 | 20 | 106 | 6 | 0.025 | Node L | |
| A-12 | 20 | 106 | 6 | 0.025 | | Node M |
| A-13 | 20 | 106 | 6 | 0.025 | Node N | |
| A-14 | 20 | 106 | 6 | 0.025 | | Node O |
| A-15 | 20 | 106 | 6 | 0.025 | Node P | |
| A-16 | 20 | 106 | 6 | 0.025 | | Node Q |
| A-17 | 20 | 106 | 6 | 0.025 | Node R | |
| A-18 | 20 | 106 | 6 | 0.025 | | Node S |
| A-19 | 20 | 106 | 6 | 0.025 | Node T | |
| A-20 | 20 | 106 | 6 | 0.025 | | |

**Table 2. Summary of leakage conditions.**

| ID | Leak aperture | Gas supply by user 1 | Gas supply pressure of user 1 | Gas supply form | Leak hole form |
|----|---------------|----------------------|-------------------------------|-----------------|----------------|
|    | (mm)          | (m$^3$/h)            | (kPa)                         |                 |                |
| 1  | 15            | 1000                 | 200                           | constant        | circle         |
| 2  | 15            | 2000                 | 200                           | constant        | circle         |
| 3  | 20            | 1000                 | 200                           | constant        | circle         |
| 4  | 15            | 1000                 | 400                           | constant        | circle         |
| 5  | 15            | 1000                 | 200                           | constant        | triangle       |
| 6  | 15            | 1000                 | 200                           | non-constant    | circle         |

and non-constant gas supply), and the shape and size of the leak hole. In order to quantitatively describe the influence of various factors on the gas flow at the user end, six leakage conditions are designed based on the idea of control variables (Table 2). Among them, the parameters of working condition 1 are all basic settings, and working condition 1 is taken as the basic working condition. A control condition established based on the value of a certain effective factor. In addition, the location of the leak point will also affect the gas flow on the user side. The official website model constructed in the subsection 'Construct a pipe network model' includes 19 nodes. This article assumes that leakage occurs at the node. For all leakage conditions, leakage point 3 needs to be simulated from node B to node T. Therefore, the simulation needs to be executed 114 times. At the same time, if there is no leakage, the working pressure of the control gas source user is fixed. After the leakage, keep the working pressure of the load user 2 constant; change the gas supply parameters of the gas source user 1 according to the actual situation to realize the leakage simulation under the constant pressure state on the user side. After the leakage occurs, the working pressure of load user 2 should be kept constant, and the gas supply parameters of the gas source user 1 should be changed according to the actual situation to realize the leakage simulation side under the constant pressure state of the user.

Before the simulation, the value range of the relevant variables and the simulation rules are set as follows: the air pressure and flow at user 1 do not exceed the set upper limit, the supply volume is fixed, and the pressure does not exceed 400k Pa. The boundary condition of load user 2 is the minimum pressure, and the supply pressure of air source user 1 is fixed. And the leakage coefficient is 1, and the external pressure is standard atmospheric pressure.

One round of simulation lasts for 180 s, and the results are output every second. A total of 180 observations were obtained after simulation. During the first 60 s of the simulation, the normal air supply condition without leakage was simulated. Simulate the leakage conditions in 61–180 s, and set the 61st s as the first of the leakage time T. According to the designed leakage conditions, change the relevant parameters of leak point 3 to perform a new round of simulation.

## Result and analysis

### Analysis of simulation results under different working conditions

It can be seen from the simulation results that the changes in user-side flow are mainly concentrated within 28s after the leak occurs. During this period, the corresponding changes in user-side flow under six operating conditions were plotted (Fig 2). In each sub-graph, the user-side flow changes under this working condition are presented in turn, and the leakage points are set at 19 nodes.

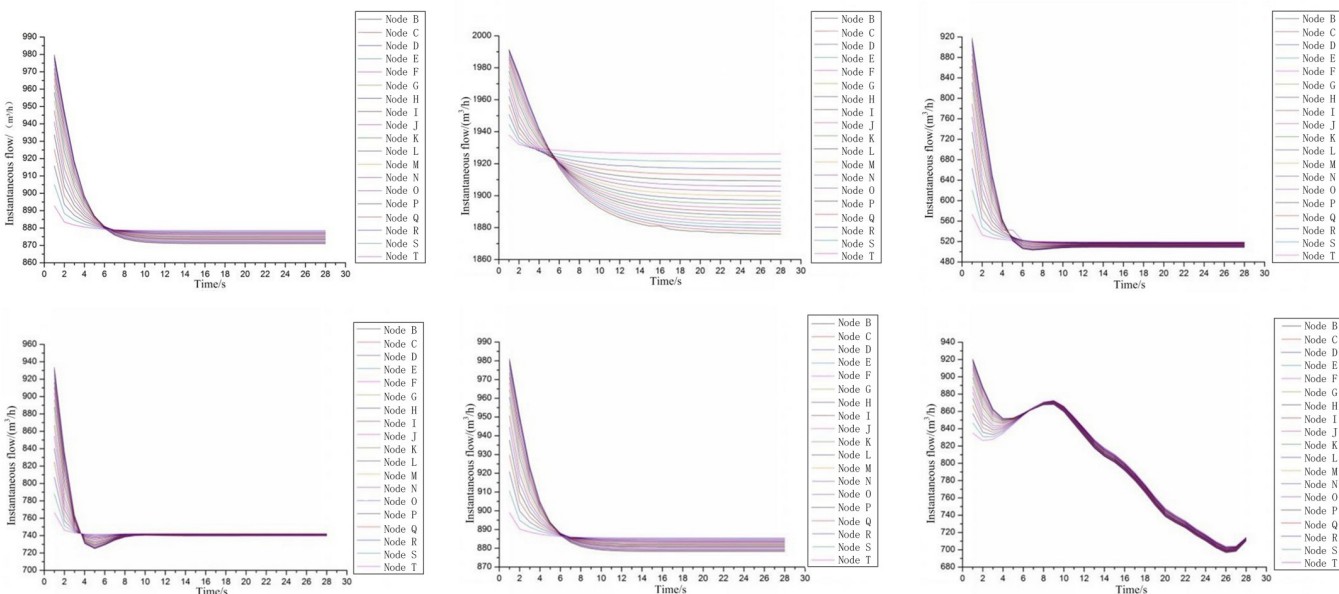

**Fig 2. The user side flow rate changes by second under leakage condition.** (a) Under leak condition 1, (b) Under leak condition 2, (c) Under leak condition 3, (d) Under leak condition 4, (e) Under leak condition 5, (f) Under leak condition 6.

1. Through the horizontal comparison of the sub-graphs in Fig 2A–2E, it is found that the instantaneous flow rate on the user side shows a trend of decreasing and tending to be stable. As the air supply mode of working condition 6 is discontinuous, the instantaneous flow of user side is in constant fluctuation in Fig 2F. And the sub-graph(f) shows that when the leakage occurred at different nodes, the instantaneous flow on the user side returned to a consistent level about 6 seconds after the leakage occurred, and then showed similar fluctuation characteristics, which is related to the gas supply mode.

2. The instantaneous flow rate on the user side after leakage fluctuates no more than 20s according to Fig 2. And the fluctuate duration under each working condition is 10s, 20s, 10s, 9s, 10s, 6s respectively. In the study of Wang Wei et al., the influence of leakage diameter and location on the duration of flow fluctuation caused by leakage was discussed, and the conclusion was drawn that no matter how big the leakage aperture is and where the leakage occurs, the flow needed about 8 s from fluctuation to stability. In this paper, the leakage diameter of working conditions 1 and 3 in this paper is 15mm and 20mm respectively, and the parameters of other working conditions are consistent. The fluctuation duration of user flow in these two working conditions is about 10s, and the fluctuation duration of flow at different leakage points is basically the same (see Fig 2A and 2(C)), which is similar to the conclusion of Wang Wei. This illustrates the rationality and robustness of the simulation results.

3. Under the same working conditions, when the leakage points are located at different nodes, the leakage situation closer to the gas source always takes longer to reach a stable state. Moreover, the initial value of the change and the range of change are also very different. When the leakage point is close to the gas source point, compared with other leakage situations, the initial value of the instantaneous flow rate on the user side is larger, but the rate of decrease is faster and the rate of decrease is larger. After stabilization, the flow value becomes smaller.

4. It can be seen from Fig 2B that under this working condition, when the leakage point is located at different nodes, there is a significant difference between the fluctuation duration of the user side flow and the steady flow value. The closer the leak point is to the gas source point, the longer it takes to reach equilibrium and the smaller the steady flow value. However, this phenomenon is not reflected in other working conditions, indicating that it is greatly affected by the flow rate of the gas source point.

5. It can be seen from Fig 2D that under this working condition, if the gas flow-pressure increases to 400kPa, the location of the leak point has little effect on the steady flow on the user side. After reaching this working condition, the flow curves highly overlap. stable state. When the pressure is 200kPa, the steady flow on the user side is between [870,880] (m$^3$/h). When the pressure increases to 400kPa, the steady flow drops to 740m$^3$/h. It can be seen that reducing the gas delivery pressure within a reasonable range can effectively reduce the gas loss after leakage.

## Quantitative analysis of the relationship between user-side flow and leakage location

**The advisable monitoring period.**    According to the analysis in the subsection 'Analysis of simulation results under different working conditions', the user-side flow fluctuation caused by the leakage will return to a stable state within 20 seconds. Therefore, determining the flow fluctuation period helps to accurately capture the flow change process on the user side and locate the leakage point. It can be seen from Fig 2 that under different leakage conditions, the duration of flow fluctuations monitored by the user side is 10-20s, and the abnormal flow changes caused by the leakage location are different, but the time cost of stable leakage is basically the same. Therefore, the critical time for leak detection and positioning using abnormal fluctuations on the user side is 20s. If the leak takes a long time and the flow has reached a steady state, it is difficult to use flow disturbance data for leak detection and location. Therefore, the advisable monitoring period could be determined as20 seconds to detect a complete flow fluctuation process. Moreover, it is noted that in the five leakage conditions shown in Fig 2A, 2C–2F, the fluctuation duration of user-side flow is within 10s, and the leakage conditions shown in (b) also show significant flow fluctuation within 10s. Therefore, if the monitoring period is set to 10s, the vast majority of leak conditions can be detected and the detection efficiency is improved, which is a better choice for both detection rate and detection efficiency. This conclusion is similar to that of Wang Kai et al.

**The flow fluctuating regulation of the user side.**    According to the simulation results and the best monitoring period determined in the subsection 'The advisable monitoring period', the flow disturbance amplitude on the user side within 8s after the leakage is selected to draw the curve. The ordinate is the disturbance amplitude, and the calculation method is the ratio of the flow rate difference between two adjacent seconds to the flow rate in the previous second (see Formula 5).

$$R_{disturbing} = \frac{(flow_t - flow_{t-1})}{flow_{t-1}} \tag{5}$$

Where $flow_t$, $flow_{t-1}$ refer to user side flow at time t and t-1 respectively.

Fig 3 shows the flow disturbance amplitude of user-end caused by different leakage locations within 8S after the leakage occurs under condition 1. It can be seen from Fig 3 that when the leakage point is located from node B to I (the first 40% of the relative distance from the air source), the airflow disturbance amplitude curve first increases and then decreases. When the

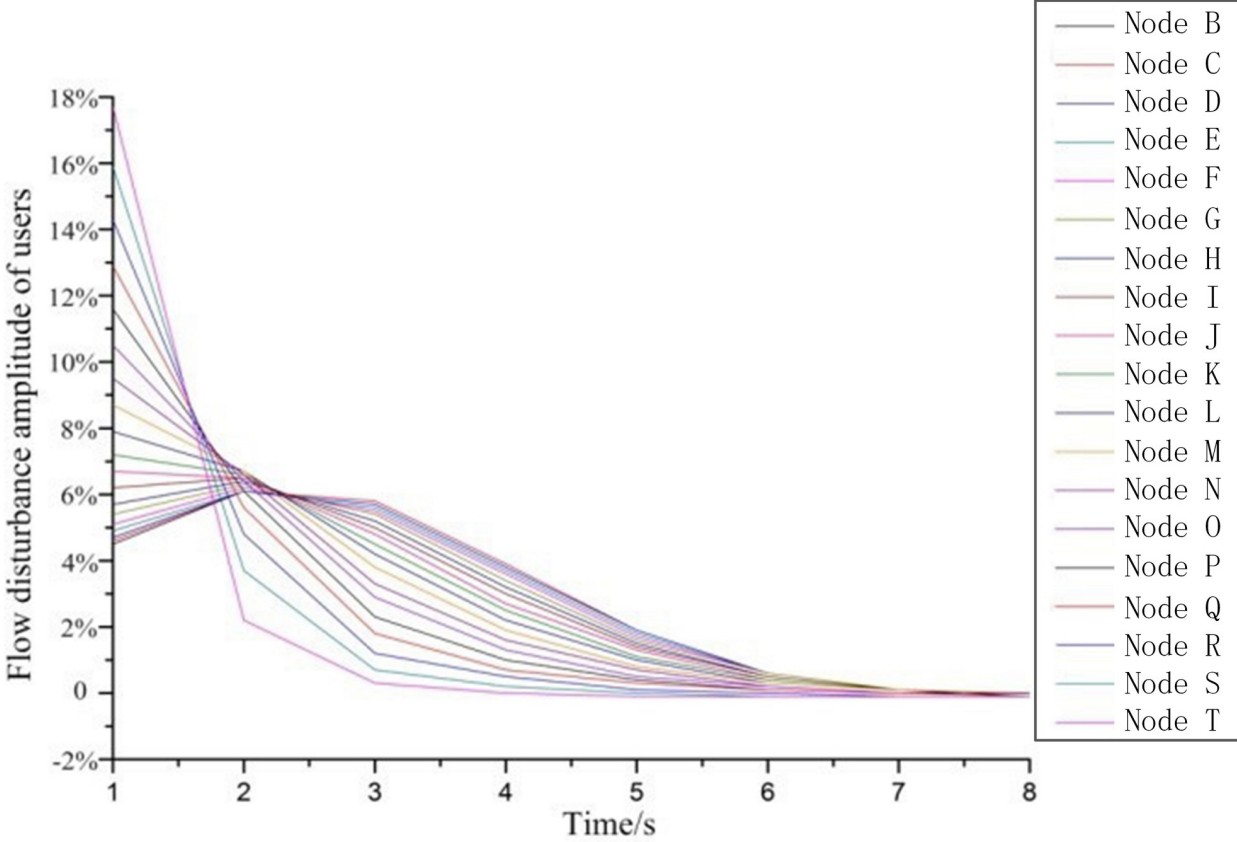

**Fig 3. The user's per second flow disturbance amplitude when leakage occurs at different nodes after fitting.**

leakage is at the J-T junction (the last 60% of the relative distance from the gas source), the flow disturbance amplitude curve continues to decrease. On the other hand, the closer the leakage point is to the user, the greater the flow fluctuations, and the further away from the user, the smaller the flow fluctuations.

## Conclusion

1. It is found in this paper that when the middle and low-pressure gas pipeline leaks, the user-side flow changes from fluctuating to stable and shows a certain change cycle and law, which is related to the location of the leak. Based on this, we propose a method to determine the location of the leak. In addition, this method can detect and locate leakage using abnormal fluctuations on the user-side.

2. Besides the location of the pipeline leak, there are many other influencing factors, such as air supply pressure and flow, and the size of the leak. It is found that the higher the air supply pressure, the larger the air supply flow, the smaller the leakage aperture, and the longer the duration of flow fluctuations, but they can all recover from fluctuations to a stable state within 10s. Therefore, the best monitoring period for user flow is set to 10s.

3. The location of the leak can be roughly determined based on the abnormal flow fluctuations within 10s after the leak occurs. When the leakage is in the first 40% of the relative distance

from the gas source, the abnormal flow fluctuation first increases and then decreases, otherwise, the flow decreases monotonously.

4. The location of the leak can be carefully judged based on the magnitude of abnormal flow fluctuation in the flow rate for 10 consecutive seconds after the leak occurred, due to the closer the leakage point is to the user side, the more significant the abnormal flow fluctuation will be.

## Discussion

This research can provide a basis for the detection and location of the leakage of medium and low-pressure gas pipelines. It can be expected that the method proposed can be embedded in the intelligent analysis system of the medium and low-pressure gas pipeline network, changing the leakage detection method of manual inspection. The detection pattern of medium and low-pressure gas leakage accidents will be changed from passive notification to active monitoring.

The simulation method proposed in this paper still has the following deficiencies:

1. There are some mechanical errors in user-side pressure and flow measurement that affect the leak detection accuracy of the theoretical formula directly. Domestic regulations allow measuring equipment to have a measurement error of 1% to 3% [23], but the structural defects of the instrument will further enlarge the measurement error as the service life of measuring equipment increases.

2. Restrictions on pipeline laying conditions. Most of the outdoor gas pipelines in our country are laid underground. These pipelines must consider the soil pore leakage rate [24]. The theoretical formula adopted in this paper is simplified, which defaults to the natural gas pipelines as overhead pipelines. And the relevant soil pore leakage rate is set to 1 [25], this indicator could be rectified in further research.

3. It can be considered to calculate multiple parameters (pressure difference, flow difference, and temperature difference) under the same working condition simultaneously and realize the method of identifying leakage working conditions based on multivariate data fusion.

4. The assumption that the gas is an ideal gas and the flow process is isothermal flow will lead to a certain error between the simulation results and the actual situation. The influence of gas type and temperature on the change of user flow rate should be considered in further study to make the simulation results closer to reality.

## Supporting information

**S1 Dataset.**
(XLSX)

## Author Contributions

**Conceptualization:** Xiaomin Wang, Zhengshan Luo.

**Data curation:** Qingqing Wang.

**Formal analysis:** Yulei Kong.

**Funding acquisition:** Zhengshan Luo.

**Investigation:** Yulei Kong, Qingqing Wang.

**Methodology:** Xiaomin Wang.

**Supervision:** Xiaomin Wang.

**Writing – review & editing:** Xiaomin Wang.

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
