## [Decision Letter · Decision Letter 0]

28 Feb 2022

PONE-D-22-03731Based on abnormal fluctuations in user-side flow Simulation analysis of low- and medium-pressure gas pipeline leakage monitoringPLOS ONE

Dear Dr. Wang,

Thank you for submitting your manuscript to PLOS ONE. After careful consideration, we feel that it has merit but does not fully meet PLOS ONE’s publication criteria as it currently stands. Therefore, we invite you to submit a revised version of the manuscript that addresses the points raised during the review process.

We look forward to receiving your revised manuscript.

Kind regards,

Feng Chen

Academic Editor

PLOS ONE

Journal Requirements:

3. We note you have included a table to which you do not refer in the text of your manuscript. Please ensure that you refer to Table 3 in your text; if accepted, production will need this reference to link the reader to the Table.

Reviewers' comments:

Reviewer's Responses to Questions

**Comments to the Author**

1. Is the manuscript technically sound, and do the data support the conclusions?

Reviewer #1: Yes

Reviewer #2: Yes

2. Has the statistical analysis been performed appropriately and rigorously? 

Reviewer #1: Yes

Reviewer #2: Yes

3. Have the authors made all data underlying the findings in their manuscript fully available?

Reviewer #1: Yes

Reviewer #2: Yes

4. Is the manuscript presented in an intelligible fashion and written in standard English?

Reviewer #1: Yes

Reviewer #2: Yes

5. Review Comments to the Author

Reviewer #1: Review comments:

1) The leakage detection method of medium and low pressure gas pipeline based on flow difference has been widely used. The author can reflect the innovation by modifying the introduction part.

2) In this paper, pipeline Studio software is used to simulate 6 groups of leakage conditions, and the simulation description is clear. The author can calculate multiple parameters (pressure difference, flow difference and temperature difference) under the same working condition at the same time, and realize the method of identifying leakage working condition based on multivariate data fusion.

3) This paper use the numerical simulation method to simulate the leakage dynamic calculation of medium and low-pressure urban gas pipeline, and the single numerical simulation calculation result is uncertain. Fig. 1 and Table 1 give a specific and detailed numerical calculation model. The author can carry out comparative experiments with this model to compare and verify the accuracy and robustness of the simulation results, or find references to compare and explain the simulation results.

4) In the parameter setting, for the gas transmission pipeline with the inner diameter of 106mm, whether the leakage hole setting of 15-20mm is reasonable, and whether it can be considered that this method can only be applied to the case of sudden large leakage.

5) For Fig. 2, I don't know what the letters from B to t mean. It is not explained in the legend and reflected in the text. It is suggested to supplement.

Recommend employment after modification.

Reviewer #2: This work aims at the monitoring of low- and medium-pressure gas pipeline leakage by analyzing abnormal fluctuations of user-side flow. This paper is attractive to practitioners in the pipeline industry. Although the research results of this paper are generally correct, some conditions and conclusions are not well combined with practice. Therefore, this article needs to be slightly modified.

Major comments:

1. Page 6, line 9-12: The assumptions that the gas is an ideal gas and the flow process is isothermal flow, will lead to a certain error between the simulation results and the actual situation.

2. Page 7, line 20-22: It can be seen from Figure 2 that the assertion that "the time required for this trend to reach stability under various working conditions is 6s, 14s, 6s, 8s, 8s and 6s respectively" is not correct. It is recommended to carefully refer to figure 2 Determine the time required for this trend to reach stability under various working conditions.

3. Page 9, line 3-10: According to figure 2, not all flow fluctuations on the user side caused by leakage conditions return to a stable state within 10 seconds. The duration of flow fluctuations on the user side caused by the first five leakage conditions is 10-20s, and the flow fluctuations on the user side caused by the sixth leakage condition do not tend to be stable within the displayed time interval. The optimal monitoring period shall be determined according to the actual time to reach stability

4. Page 10, line 1-7: (1) It is not explained that the calculation of flow disturbance amplitude in Figure 3 is based on the flow data of which of the six leakage conditions.

(2) From Figure 3, the conclusion that "no matter where the leakage is in the pipeline, the flow disturbance amplitude tends to be consistent within 2 ~ 3 seconds after the leakage" is not accurate. For example, when the leakage occurs at node T, the flow disturbance amplitude is 2% in the second second and tends to 0 in the third second. It is suggested to modify or delete this clause.

6. PLOS authors have the option to publish the peer review history of their article (what does this mean?). If published, this will include your full peer review and any attached files.

Reviewer #1: No

Reviewer #2: No

---

## [Author Response · Author response to Decision Letter 0]

9 Apr 2022

Reviewer #1: Review comments:

1) The leakage detection method of medium and low-pressure gas pipelines based on flow difference has been widely used. The author can reflect the innovation by modifying the introduction part.

Re: “Compared to the leak characteristics, the user side pressure and flow data are easier to monitor, and there are few studies on pipeline leak detection and location based on user side flow data yet. Therefore, this paper intends to detect and locate gas pipeline leak by using the abnormal fluctuation of flow on the user side generated at the moment of middle and low-pressure gas pipeline leak combined with other detection technologies under different working conditions, which provides an active basis theory, method, and conclusion for the low-pressure pipeline leak detection and location work.” 

2) In this paper, pipeline Studio software is used to simulate 6 groups of leakage conditions, and the simulation description is clear. The author can calculate multiple parameters (pressure difference, flow difference and temperature difference) under the same working condition at the same time, and realize the method of identifying leakage working condition based on multivariate data fusion.

Re: Thanks to the suggestions of the reviewer, it is a great idea to conduct leak detection through the multi-source data fusion method. The purpose of this paper is to explore a convenient and fast leak detection method based on the availability of pipeline flow data on the user side. The experiment has not yet included temperature and pressure measurements at each monitoring point. Therefore, we put the suggestions of reviewers in the "Discussion" section as a starting point for further study. 

3) This paper uses the numerical simulation method to simulate the leakage dynamic calculation of medium and low-pressure urban gas pipeline, and the single numerical simulation calculation result is uncertain. Fig. 1 and Table 1 give a specific and detailed numerical calculation model. The author can carry out comparative experiments with this model to compare and verify the accuracy and robustness of the simulation results, or find references to compare and explain the simulation results.

Re: Thank the reviewer for pointing out the shortcomings of the experimental analysis， we’ve quoted references to compare and explain the simulation results. 

“In the study of Wang Wei et al., the influence of leakage diameter and location on the duration of flow fluctuation caused by leakage was discussed, and the conclusion was drawn that no matter how big the leakage aperture is and where the leakage occurs, the flow needed about 8 s from fluctuation to stability. In this paper, the leakage diameter of working conditions 1 and 3 in this paper is 15mm and 20mm respectively, and the parameters of other working conditions are consistent. The fluctuation duration of user flow in these two working conditions is about 10s, and the fluctuation duration of flow at different leakage points is basically the same (see Fig 2. (a) and (c)), which is similar to the conclusion of Wang Wei. This illustrates the rationality and robustness of the simulation results.” 

4) In the parameter setting, for the gas transmission pipeline with the inner diameter of 106mm, whether the leakage hole setting of 15-20mm is reasonable, and whether it can be considered that this method can only be applied to the case of sudden large leakage.

Re: Thanks for the suggestions, this paper didn’t elaborate on the applicable scenario of the proposed method before. The method in this paper aims to quickly find the flow anomaly caused by leakage through short-term flow monitoring, and then locate the leakage location. This method has a short monitoring period (set to 10s in this paper) and strong real-time performance, which is suitable for sudden leakage scenarios. According to the analysis results in this paper, the fluctuation caused by the leakage with a larger leak size lasts for a longer duration and can be completely captured in a short monitoring period. Therefore, the method in this paper is suitable for the case of sudden large pore size leakage.

5) For Fig. 2, I don't know what the letters from B to t mean. It is not explained in the legend and reflected in the text. It is suggested to supplement.

Re: We are very sorry for the shortcomings in our drawings. B to T in the legend of Figure 2 represent the pipe nodes, where the leak point may occur. We have now modified the relevant images (Figures 2 and 3).

(a) Under leak condition 1 (b) Under leak condition 2

(c) Under leak condition 3 (d) Under leak condition 4

(e) Under leak condition 5 (f) Under leak condition 6

Figure 2. The user side flow rate changes by second under leakage condition

Fig.3 The user's per second flow disturbance amplitude when leakage occurs at different nodes after fitting

Reviewer #2: Review comments:

This work aims at the monitoring of low- and medium-pressure gas pipeline leakage by analyzing abnormal fluctuations of user-side flow. This paper is attractive to practitioners in the pipeline industry. Although the research results of this paper are generally correct, some conditions and conclusions are not well combined with practice. Therefore, this article needs to be slightly modified.

Major comments:

1. Page 6, line 9-12: The assumptions that the gas is an ideal gas and the flow process is isothermal flow, will lead to a certain error between the simulation results and the actual situation.

Re: Thanks to the reviewer for pointing out the problem. We did not address this issue clearly in our previous manuscript. Under the assumption of "ideal gas and isothermal flow", the simulation results are indeed different from the actual situation. However, considering that the simulation of all working conditions is based on this assumption, it can meet the research demand of locating the leakage point based on the flow change rule after leakage. At the same time, we believe that the suggestions of reviewers are of great significance for improving our research. We will optimize the experimental design in the follow-up research to make the simulation results closer to reality.

2. Page 7, line 20-22: It can be seen from Figure 2 that the assertion that "the time required for this trend to reach stability under various working conditions is 6s, 14s, 6s, 8s, 8s and 6s respectively" is not correct. It is recommended to carefully refer to figure 2 Determine the time required for this trend to reach stability under various working conditions.

Re: Thank the reviewer for pointing out the problems in the analysis of Figure 2. After we read the figures carefully, we modified the relevant statement as follows.

“(2) The instantaneous flow rate on the user side after leakage fluctuates no more than 20s according to Fig. 2. And the fluctuate duration under each working condition is 10s, 20s, 10s, 9s, 10s, 6s respectively.”

3. Page 9, line 3-10: According to figure 2, not all flow fluctuations on the user side caused by leakage conditions return to a stable state within 10 seconds. The duration of flow fluctuations on the user side caused by the first five leakage conditions is 10-20s, and the flow fluctuations on the user side caused by the sixth leakage condition do not tend to be stable within the displayed time interval. The optimal monitoring period shall be determined according to the actual time to reach stability.

Re: As for the doubts pointed out by reviewers about the continuous fluctuation of user-side flow in the sixth working condition, the main reason is that the gas supply mode in this working condition is intermittent, which is different from the continuous gas supply in other working conditions. Under this condition, even without leakage, the flow is still unstable. For several working conditions, the expression of gas flow fluctuation after leakage is modified as follows:

“(1) Through the horizontal comparison of the subgraphs (a)~(e) in Fig. 2, it is found that the instantaneous flow rate on the user side shows a trend of decreasing and tending to be stable. As the air supply mode of working condition 6 is discontinuous, the instantaneous flow of the user side is in constant fluctuation in Fig. 2(f). And the subgraph (f) shows that when the leakage occurred at different nodes, the instantaneous flow on the user side returned to a consistent level about 6 seconds after the leakage occurred, and then showed similar fluctuation characteristics, which is related to the gas supply mode.”

4. Page 10, line 1-7: (1) It is not explained that the calculation of flow disturbance amplitude in Figure 3 is based on the flow data of which of the six leakage conditions.

(2) From Figure 3, the conclusion that "no matter where the leakage is in the pipeline, the flow disturbance amplitude tends to be consistent within 2 ~ 3 seconds after the leakage" is not accurate. For example, when the leakage occurs at node T, the flow disturbance amplitude is 2% in the second second and tends to 0 in the third second. It is suggested to modify or delete this clause.

Re: （1）Thanks to reviewers for pointing out the problem. We do not make clear what the line chart in Figure 3 shows. The interpretation is as follows: Fig.3 does not target a specific working condition but fits the flow data of a leakage point under different working conditions into a curve, and the flow fluctuation curves of 19 leakage points are plotted in Fig.3.

（2）The amplitude of user-side flow disturbance occurring at the same leak point under different working conditions is fitted into a curve. The fitting curves at 19 leakage points are shown in FIG. 3.

---

## [Decision Letter · Decision Letter 1]

12 May 2022

PONE-D-22-03731R1Based on abnormal fluctuations in user-side flow Simulation analysis of low- and medium-pressure gas pipeline leakage monitoringPLOS ONE

Dear Dr. Wang,

Thank you for submitting your manuscript to PLOS ONE. After careful consideration, we feel that it has merit but does not fully meet PLOS ONE’s publication criteria as it currently stands. Therefore, we invite you to submit a revised version of the manuscript that addresses the points raised during the review process.

We look forward to receiving your revised manuscript.

Kind regards,

Feng Chen

Academic Editor

PLOS ONE

Journal Requirements:

Reviewers' comments:

Reviewer's Responses to Questions

**Comments to the Author**

1. If the authors have adequately addressed your comments raised in a previous round of review and you feel that this manuscript is now acceptable for publication, you may indicate that here to bypass the “Comments to the Author” section, enter your conflict of interest statement in the “Confidential to Editor” section, and submit your "Accept" recommendation.

Reviewer #1: All comments have been addressed

Reviewer #2: All comments have been addressed

2. Is the manuscript technically sound, and do the data support the conclusions?

Reviewer #1: Yes

Reviewer #2: Yes

3. Has the statistical analysis been performed appropriately and rigorously? 

Reviewer #1: Yes

Reviewer #2: Yes

4. Have the authors made all data underlying the findings in their manuscript fully available?

Reviewer #1: No

Reviewer #2: Yes

5. Is the manuscript presented in an intelligible fashion and written in standard English?

Reviewer #1: Yes

Reviewer #2: Yes

6. Review Comments to the Author

Reviewer #1: The last part of the paper (conclusion and prospect) are too complicated. It is suggested to summarize and highlight the innovation.

Reviewer #2: Thank author for your serious reply to my comments. These changes you have made can make this article meet the publishing requirements. However, if you can make a precise explanation of which working condition the data in Figure 3 comes from, it will make the article more rigorous.

7. PLOS authors have the option to publish the peer review history of their article (what does this mean?). If published, this will include your full peer review and any attached files.

Reviewer #1: **Yes: **Shaohua Dong

Reviewer #2: No

---

## [Author Response · Author response to Decision Letter 1]

24 May 2022

Reviewer #1: Review comments:

1. The last part of the paper (conclusion and prospect) are too complicated. It is suggested to summarize and highlight the innovation.

Re: Thanks to the reviewer for the suggestion about simplifying the conclusion and highlighting the innovation. We’ve revised the conclusion and prospect carefully as follows.

“Conclusion

(1) It is found in this paper that when the middle and low-pressure gas pipeline leaks, the user-side flow changes from fluctuating to stable and shows a certain change cycle and law, which is related to the location of the leak. Based on this, we propose a method to determine the location of the leak. In addition, this method can detect and locate leakage using abnormal fluctuations on the user-side.

(2) Besides the location of the pipeline leak, there are many other influencing factors, such as air supply pressure and flow, and the size of the leak. It is found that the higher the air supply pressure, the larger the air supply flow, the smaller the leakage aperture, and the longer the duration of flow fluctuations, but they can all recover from fluctuations to a stable state within 10s. Therefore, the best monitoring period for user flow is set to 10s.

(3) The location of the leak can be roughly determined based on the abnormal flow fluctuations within 10s after the leak occurs. When the leakage is in the first 40% of the relative distance from the gas source, the abnormal flow fluctuation first increases and then decreases, otherwise, the flow decreases monotonously. 

(4) The location of the leak can be carefully judged based on the magnitude of abnormal flow fluctuation in the flow rate for 10 consecutive seconds after the leak occurred, due to the closer the leakage point is to the user side, the more significant the abnormal flow fluctuation will be.

Discussion

This research can provide a basis for the detection and location of the leakage of medium and low-pressure gas pipelines. It can be expected that the method proposed can be embedded in the intelligent analysis system of the medium and low-pressure gas pipeline network, changing the leakage detection method of manual inspection. The detection pattern of medium and low-pressure gas leakage accidents will be changed from passive notification to active monitoring.

The simulation method proposed in this paper still has the following deficiencies：

(1) There are some mechanical errors in user-sidepressure and flow measurement that affect the leak detection accuracy of the theoretical formula directly. Domestic regulations allow measuring equipment to have a measurement error of 1% to 3% [23], but the structural defects of the instrument will further enlarge the measurement error as the service life of measuring equipment increases.

(2) Restrictions on pipeline laying conditions. Most of the outdoor gas pipelines in our country are laid underground. These pipelines must consider the soil pore leakage rate [24]. The theoretical formula adopted in this paper is simplified, which defaults to the natural gas pipelines asoverhead pipelines. And the relevant soil pore leakage rate is set to 1 [25], this indicator could be rectified in further research.

(3) It can be considered to calculate multiple parameters (pressure difference, flow difference, and temperature difference) under the same working condition simultaneously and realize the method of identifying leakage working conditions based on multivariate data fusion.

(4) The assumption that the gas is an ideal gas and the flow process is isothermal flow will lead to a certain error between the simulation results and the actual situation. The influence of gas type and temperature on the change of user flow rate should be considered in further study to make the simulation results closer to reality.

”

Reviewer #2: Review comments:

2. Thank author for your serious reply to my comments. These changes you have made can make this article meet the publishing requirements. However, if you can make a precise explanation of which working condition the data in Figure 3 comes from, it will make the article more rigorous.

Re: Thanks to the reviewer for this suggestion to improve our work, our description of the case in Figure 3 was not clear before and is explained below. Figure 3 shows the flow disturbance amplitude of user-end caused by different leakage locations within 8S after the leakage occurs under condition 1. Condition 1 is the basic condition of the simulation experiments in this paper, and all leakage parameters are the basic settings (see Table 2). It can be seen in Figure 3 that the flow fluctuation of node T returns to 0 first, that is, the instantaneous flow reaches a balance, and the fluctuation range of node B is the last to return to 0. This rule can be verified from Figure 2. It can be seen from Figure 2 that under any working condition, the changes of user-end flow at different leak locations show similar laws: when the leak location is at the T-node close to the user-end, the flow first reaches a balanced state. The farther the leak location is from the user-end, the longer it takes to reach equilibrium. When the leak occurs at node B, it takes the longest time for the flow to reach the equilibrium state.

---

## [Editor Report · Decision Letter 2]

8 Jun 2022

Based on abnormal fluctuations in user-side flow Simulation analysis of low- and medium-pressure gas pipeline leakage monitoring

PONE-D-22-03731R2

Dear Dr. Wang,

We’re pleased to inform you that your manuscript has been judged scientifically suitable for publication and will be formally accepted for publication once it meets all outstanding technical requirements.

Kind regards,

Feng Chen

Academic Editor

PLOS ONE
---

## [Editor Report · Acceptance letter]

15 Jun 2022

PONE-D-22-03731R2 

Based on abnormal fluctuations in user-side flow Simulation analysis of low- and medium-pressure gas pipeline leakage monitoring 

Dear Dr. Wang:

I'm pleased to inform you that your manuscript has been deemed suitable for publication in PLOS ONE. Congratulations! Your manuscript is now with our production department. 

Kind regards, 

on behalf of

Dr. Feng Chen 

Academic Editor

PLOS ONE